# Fisher-Information-Matrix-Based USBL Cooperative Location in USV–AUV Networks

**DOI:** 10.3390/s23177429

**Published:** 2023-08-25

**Authors:** Ziyuan Wang, Jingzehua Xu, Yuanzhe Feng, Yijing Wang, Guanwen Xie, Xiangwang Hou, Wei Men, Yong Ren

**Affiliations:** 1Department of Electronic Engineering, Tsinghua University, Beijing 100084, China; wangziyu21@mails.tsinghua.edu.cn (Z.W.); hxw21@mails.tsinghua.edu.cn (X.H.); reny@tsinghua.edu.cn (Y.R.); 2Tsinghua Shenzhen International Graduate School, Tsinghua University, Shenzhen 518055, China; 19955778426@163.com; 3Ocean College, Zhejiang University, Zhoushan 316021, China; 3200100040@zju.edu.cn (Y.F.); 3200101418@zju.edu.cn (G.X.); 4Department of Computer Science and Technology, Xidian University, Xi’an 710126, China; xidianwyj@126.com

**Keywords:** USV–AUV network, USBL acoustic cooperative location with orthogonal array, Fisher information matrix, Dubins path planning functions

## Abstract

With the development of underwater technology and the increasing demand for ocean development, more and more intelligent equipment is being applied to underwater scientific missions. Specifically, autonomous underwater vehicle (AUV) clusters are being used for their flexibility and the advantages of carrying communication and detection units, often performing underwater tasks in formation. In order to locate AUVs with high precision, we introduce an unmanned surface vehicle (USV) with global positioning system (GPS) and propose a USV–AUV network. Furthermore, we propose an ultra-short baseline (USBL) acoustic cooperative location scheme with an orthogonal array, which is based on underwater communication with sonar. Based on the derivation of the Fisher information matrix formula under Cartesian parameters, we analyze the positioning accuracy of AUVs in different positions under the USBL positioning mode to derive the optimal array of the AUV formation. In addition, we propose a USV path planning scheme based on Dubins path planning functions to assist in locating the AUV formation. The simulation results verify that the proposed scheme can ensure the positioning accuracy of the AUV formation and help underwater research missions.

## 1. Introduction

With the rapid development of scientific research in the marine field, tools for exploring the ocean are emerging in an endless stream. Underwater robotic technology has attracted wide attention due to its potential applications in various fields such as environmental monitoring, seabed exploration and biological investigation [1,2]. The success of underwater robotic missions depends to a large extent on the accurate and reliable positioning of these robotic systems [3]. Underwater robots need to complete various tasks, and precise control of the robot’s working area can be achieved through positioning to improve the efficiency of the operation [4]. In an underwater environment, any problem encountered by the robot may lead to damage or loss of control. Through positioning, the position and state of the robot can be determined in real time, thereby increasing the safety of the operation and avoiding damage or accidents [5].

Currently, methods for positioning underwater robots can be mainly divided into the following categories: global positioning systems (GPSs), inertial navigation system positioning, and ultrasonic positioning. GPS determines the global position of the robot through its satellite positioning system, but underwater GPS signal transmission is susceptible to the influence of water refraction and scattering, transmission attenuation, delay and other problems, thus affecting the accuracy and efficiency of underwater positioning [6,7]. The inertial navigation positioning of an underwater vehicle uses devices such as accelerometers, gyroscopes and odometers to measure the robot’s motion and calculate its position, but long-term operation is prone to the accumulation of errors [8]. In addition, underwater robots also face some unique challenges in positioning [9], such as underwater signal attenuation, multi-path effect, and environmental noise [10,11]. Poor underwater illumination and visibility easily affect the accuracy and efficacy of robot sensors [12,13].

Fortunately, the emergence of co-localization provides a new way to solve this problem. Bahr et al. proposed a distributed algorithm that uses a set of autonomous underwater vehicles (AUVs) to dynamically calculate the locally optimal position of the beacon vehicle using information obtained from the broadcast communication of the survey vehicle in [14]. Tang et al. proposed a factor graph weighted particle-assisted distributed underwater node cooperative location algorithm (WP-DUCP) and applied it to underwater networks to improve the resolution of underwater positioning in [15]. Vasilijevi et al. established the internet of underwater things (IoUT) based on the unmanned surface vehicle (USV), so as to improve the efficiency of underwater positioning in [16]. Lin et al. proposed a hierarchical positioning framework based on a USBL positioning system to locate each AUV in the network in [17] and verified that the proposed scheme performed better than the existing distributed strategies.

However, the above collaborative positioning methods all have shortcomings in complex underwater environments: on the one hand, with an increase in the number of robots and sensors, the underwater acoustic channel becomes complicated, the computational complexity increases, the underwater node load is limited, and other problems arise [18,19]; on the other hand, the system requires a large amount of energy to achieve positioning and communication, so there are high requirements for the battery life and operating costs of the equipment.

Therefore, this study proposes a new surface–underwater cooperative positioning system based on an ultra-short baseline (USBL) acoustic positioning system, which is composed of an algorithm and a simulation to locate multiple underwater AUVs considering the Dubins path planning functions for a USV. The contributions of this paper are as follows:
(1)We design a USV–AUV network structure for locating AUVs during underwater missions, where we model sonar-based underwater communication.(2)A collaborative positioning scheme based on USBL is proposed. The Fisher information matrix is used to analyze the positioning accuracy, and then the optimal array for the AUV formation is derived.(3)We design a USV path planning scheme for underwater positioning based on the Dubins paths.


The rest of this article is organized as follows: in Section 2, we introduce the system model; in Section 3, we propose a location method and introduce the Fisher information matrix to design a USV path planning scheme; in Section 4, we present the simulation experiments and a corresponding analysis of the designed schemes; finally, we give the conclusion in Section 5.

**Definitions** **1.***In this paper, the autonomous underwater vehicle is denoted as AUV, unmanned surface vehicle is denoted as USV, the internet of underwater things is denoted as IoUT, ultra-short baseline is denoted as USBL, Cramer–Rao inequality lower bound is denoted as CRLB, and global positioning system is denoted as GPS*.

## 2. System Model

### 2.1. USV–AUV Networks

We consider the application scenario shown in Figure 1. Under a certain research water area, AUVs carry out scientific research tasks at a certain depth and form a formation to move uniformly to meet the task requirements. Each AUV can move or hover at varying depths. In order to accurately obtain the position of AUVs performing underwater tasks, the surface sailing USV and the AUV formation form USV–AUV networks based on underwater acoustic communication. Specifically, the USV is wirelessly equipped with GPS and communicates with the satellite to obtain its own position information. The USV follows the AUV formation and allows each AUV to communicate with it. Then, the position of each AUV can be obtained by means of the orthogonal array sonar installed on the surface ship and the depth information returned by the AUV. Due to the harsh underwater communication environment, the AUV formation should remain within the reliable communication range $r_{c}$ of the USV, which is denoted as $\Lambda =\left\{1,\ldots ,k,\ldots ,m\right\}$. Each AUV transmits its own motion and attitude information to the USV to assist in path planning. At the same time, the AUV formation should adopt the optimal formation according to its position relative to the USV.

### 2.2. Underwater Acoustic Communication Model for USBL

The USV uses the orthogonal array USBL acoustic positioning system to measure the AUVs’ position information. To describe sonar-based communication between the USV and AUVs, we introduce the active sonar equation in an underwater environment [20]:(1)EM=SL+TS+DI−NL−DT−2TL,
where the units of the above parts are in dB. EM is the echo margin received by the USV, SL denotes the emission sound level of active sonars, TS represents the return signal strength of the AUV’s transponder, and $DI=10\log(n_{tr})$ is the directivity index of the sonar, where $n_{tr}$ is the number of transducer units. In order to analyze the transmission loss TL of acoustic signals in an underwater communication environment based on the transmission distance l and center frequency f, we introduce underwater acoustics using Throp’s model [21]:(2)$$TL=\varpi (f)\frac{l}{10^{3}}+\mathrm{lg}(l^{2}),$$
(3)$$\varpi (f)=3\times 10^{-3}+\left(2.75\times 10^{-4}\right)f^{2}+\frac{44f^{2}}{f^{2}+4100}+\frac{0.11f^{2}}{f^{2}+1},$$
where ϖ(f) is the absorption coefficient for frequency. Moreover, the underwater noise model proposed by Stojanovic is used to analyze the characteristics of high noise in an underwater communication environment [22]:(4)$$N_{tu}=10^{1.7-3\log f},$$
(5)$$N_{sh}=10^{2\varsigma +3+2.6\log(f)-6\log\left(f+0.03\right)},$$
(6)$$N_{wa}=10^{5+0.75\sqrt{w}+2\log(f)-4\log\left(f+0.4\right)},$$
(7)$$N_{th}=10^{2\log(f)-1.5},$$
where $N_{tu}$, $N_{sh}$, $N_{wa}$ and $N_{th}$ represent the ambient noise components from turbulence, ships, waves and thermal noise, respectively. Additionally, w is the wind speed measured in m/s, and ς denotes shipping activity. Thus, the noise level can be derived as
(8)$$NL=10\mathrm{lg}\left(N_{th}+N_{tu}+N_{wa}+N_{sh}\right).$$

Given the communication threshold DT of the sonar, we obtain the effective detection range of USBL:(9)$$r_{c}=\underset{l}{\arg\max}\left\{EM(l,f)\geq 0\right\}.$$

## 3. Methods

### 3.1. Orthogonal Array USBL Cooperative Location

The USBL algorithm designed in this work adopts an orthogonal array method, as shown in Figure 2, where *Xa*, *Xb*, *Ya*, *Yb* and O are 5 transducer units with identical characteristics [23]. The arrays are arranged on the USV at equal spacing and the spacing meets OXa=OXb=OYb=d/2, where d is the array spacing. A rectangular coordinate system is demonstrated in Figure 2, and P is the transponder fixed on the underwater target AUV. Determining the underwater sound speed c and the signal frequency emitted by the transducer can be used to carry out the location process. By measuring the time t from the transmitting signal by the transmitting transducer to the receiving signal by the receiving transducer, as well as the phase difference $\Delta \varphi_{x}$ between receiving units a and b on X-axis and the phase difference $\Delta \varphi_{y}$ between receiving units a and b on Y-axis, we can calculate the position parameters of each AUV:(10)$$x=\frac{c}{2\pi fd}\Delta \varphi_{x}S,$$
(11)$$y=\frac{c}{2\pi fd}\Delta \varphi_{y}S,$$
(12)$$S^{2}=\sqrt{x^{2}+y^{2}+z^{2}},$$
where S represents the length of OP, and $\left(x,y,z\right)$ represents the coordinate of the transponder P of the target AUV.

### 3.2. Fisher Information Matrix Based on USBL Location Model

The Cramer–Rao inequality lower bound (CRLB) describes the lower bound of a covariance matrix for an unbiased estimate of an unknown quantity, the reciprocal of which is the Fisher information. The Fisher information matrix is often used to measure the amount of state-variable information contained in the quantity measurement of a system [24]. The greater the Fisher information, the more information contained in the system model, the lower the uncertainty of the system, and the higher the accuracy of the state quantity. Therefore, through the analysis of Fisher information of each AUV at different positions, the positioning accuracy of different positions under the orthogonal array USBL positioning mode can be evaluated.

According to the measurement information, we can deduce Fisher information matrix formula based on Cartesian parameters. The coordinates of the USV are denoted as $\left(x,y,0\right)$, the number of AUVs to be located within the communication range is m, and the coordinates of AUV k are $\left(x_{k},y_{k},z_{k}\right)$. According to Equations (10)–(12), we can come to the following conclusion:(13)$$\Delta \varphi_{x,k}=\frac{2\pi fd(x_{k}-x)}{cS_{k}},$$
(14)$$\Delta \varphi_{y,k}=\frac{2\pi fd(y_{k}-y)}{cS_{k}},$$
(15)$$S_{k}=\sqrt{{\left(x_{k}-x\right)}^{2}+{\left(y_{k}-y\right)}^{2}+z_{k}^{2}}.$$

Since the depth of the AUV to be positioned underwater can be obtained by means of a depth sensor, the equation used to measure the positioning accuracy of the point position does not contain the depth parameter when the Fisher information matrix is established. Therefore, the measurement equation is
(16)$$Z_{k}=h_{k}(X)+u_{k},$$
where $X={\left[x,y\right]}^{T}$ is the target state vector, $h_{k}={\left[\Delta \varphi_{x,k},\Delta \varphi_{y,k}\right]}^{T}$, and $u_{k}$ is the zero mean Gaussian white noise. Let $R=\sigma^{2}I$ be the measurement noise covariance matrix, and assume that the noise covariance matrix of different target AUVs is the same. Then, the probability density function of the measured data is:(17)$$p\left(Z,X\right)=\prod_{k=1}^{m}\frac{\exp\left\{-\frac{1}{2}{\left[Z_{k}-h_{k}\left(X\right)\right]}^{T}R^{-1}\left[Z_{k}-h_{k}\left(X\right)\right]\right\}}{\sqrt{2\pi \det(R)}},$$
whose logarithmic form is
(18)$$\begin{array}{ll} \ln\left(p\left(Z,X\right)\right) & =-\frac{m\left(\ln\left(2\pi \right)+\ln\left(\det(R)\right)\right)}{2}\\ & -\frac{\sum_{k=1}^{m}{\left[Z_{k}-h_{k}\left(X\right)\right]}^{T}R^{-1}\left[Z_{k}-h_{k}\left(X\right)\right]}{2}, \end{array}$$
whose first partial derivative is
(19)$$\begin{array}{cc} \frac{\partial \ln\left(p\left(Z,X\right)\right)}{\partial x} & =\frac{1}{2}\sum_{k=1}^{m}\left\{{\left[\frac{\partial h_{k}\left(X\right)}{\partial x}\right]}^{T}R^{-1}\left[Z_{k}-h_{k}\left(X\right)\right]\right\}\\ & +\frac{1}{2}\sum_{k=1}^{m}\left\{{\left[Z_{k}-h_{k}\left(X\right)\right]}^{T}R^{-1}\frac{\partial h_{k}\left(X\right)}{\partial x}\right\}, \end{array}$$
based on which we can derive the second partial derivative as
(20)$$\begin{array}{l} \frac{\partial^{2}\ln\left(p\left(Z,X\right)\right)}{\partial x^{2}}=\frac{1}{2}\sum_{k=1}^{m}\left\{{\left[\frac{\partial^{2}h_{k}\left(X\right)}{\partial x^{2}}\right]}^{T}R^{-1}\left[Z_{k}-h_{k}\left(X\right)\right]\right\}\\ +\frac{1}{2}\sum_{k=1}^{m}\left\{{\left[Z_{k}-h_{k}\left(X\right)\right]}^{T}R^{-1}\frac{\partial^{2}h_{k}\left(X\right)}{\partial x^{2}}-2{\left[\frac{\partial h_{k}\left(X\right)}{\partial x}\right]}^{T}R^{-1}\left[\frac{\partial h_{k}\left(X\right)}{\partial x}\right]\right\}, \end{array}$$
whose expectation can be denoted as
(21)$$Ε\left[\frac{\partial^{2}\ln\left(p\left(Z,X\right)\right)}{\partial x^{2}}\right]=-\sum_{k=1}^{m}\left\{{\left[\frac{\partial h_{k}\left(X\right)}{\partial x}\right]}^{T}R^{-1}\frac{\partial h_{k}\left(X\right)}{\partial x}\right\}.$$

Similar to the above derivation, we can further conclude that:(22)$$Ε\left[\frac{\partial^{2}\ln\left(p\left(Z,X\right)\right)}{\partial x\partial y}\right]=-\sum_{k=1}^{m}\left\{{\left[\frac{\partial h_{k}\left(X\right)}{\partial x}\right]}^{T}R^{-1}\frac{\partial h_{k}\left(X\right)}{\partial y}\right\},$$
(23)$$Ε\left[\frac{\partial^{2}\ln\left(p\left(Z,X\right)\right)}{\partial y\partial x}\right]=-\sum_{k=1}^{m}\left\{{\left[\frac{\partial h_{k}\left(X\right)}{\partial y}\right]}^{T}R^{-1}\frac{\partial h_{k}\left(X\right)}{\partial x}\right\},$$
(24)$$Ε\left[\frac{\partial^{2}\ln\left(p\left(Z,X\right)\right)}{\partial y^{2}}\right]=-\sum_{k=1}^{m}\left\{{\left[\frac{\partial h_{k}\left(X\right)}{\partial y}\right]}^{T}R^{-1}\frac{\partial h_{k}\left(X\right)}{\partial y}\right\},$$
(25)$$\frac{\partial h_{k}\left(X\right)}{\partial x}=-{\left[\frac{2\pi fd\left({\left(y_{k}-y\right)}^{2}+z_{k}^{2}\right)}{cS_{k}^{3}},\frac{2\pi fd\left(x_{k}-x\right)\left(y_{k}-y\right)}{cS_{k}^{3}}\right]}^{T},$$
(26)$$\frac{\partial h_{k}\left(X\right)}{\partial y}=-{\left[\frac{2\pi fd\left(x_{k}-x\right)\left(y_{k}-y\right)}{cS_{k}^{3}},\frac{2\pi fd\left({\left(x_{k}-x\right)}^{2}+z_{k}^{2}\right)}{cS_{k}^{3}}\right]}^{T},$$
based on which we can obtain the Fisher information matrix when the surface auxiliary ship is set as the origin of coordinates:
(27)$$\begin{array}{l} J_{m}=\left[\begin{array}{c} \begin{array}{cc} -Ε\left[\frac{\partial^{2}\ln\left(p(Z,X)\right)}{\partial x^{2}}\right] & -Ε\left[\frac{\partial^{2}\ln\left(p(Z,X)\right)}{\partial x\partial y}\right] \end{array}\\ \begin{array}{cc} -Ε\left[\frac{\partial^{2}\ln\left(p(Z,X)\right)}{\partial y\partial x}\right] & -Ε\left[\frac{\partial^{2}\ln\left(p(Z,X)\right)}{\partial y^{2}}\right] \end{array} \end{array}\right]\\ =\left[\begin{array}{c} \begin{array}{cc} \sum_{k=1}^{m}\frac{4\pi^{2}f^{2}d^{2}\left({\left(S_{k}^{2}-x_{k}^{2}\right)}^{2}+x_{k}^{2}y_{k}^{2}\right)}{\sigma^{2}c^{2}S_{k}^{6}} & \sum_{k=1}^{m}\frac{4\pi^{2}f^{2}d^{2}\left({\left(S_{k}^{2}+z_{k}^{2}\right)}^{2}x_{k}y_{k}\right)}{\sigma^{2}c^{2}S_{k}^{6}} \end{array}\\ \begin{array}{cc} \sum_{k=1}^{m}\frac{4\pi^{2}f^{2}d^{2}\left({\left(S_{k}^{2}+z_{k}^{2}\right)}^{2}x_{k}y_{k}\right)}{\sigma^{2}c^{2}S_{k}^{6}} & \sum_{k=1}^{m}\frac{4\pi^{2}f^{2}d^{2}\left({\left(S_{k}^{2}-y_{k}^{2}\right)}^{2}+x_{k}^{2}y_{k}^{2}\right)}{\sigma^{2}c^{2}S_{k}^{6}} \end{array} \end{array}\right], \end{array}$$
which can be further expressed as follows when we have $\frac{y_{k}}{x_{k}}=\tan\varphi_{k}$ and $p_{k}^{2}=x_{k}^{2}+y_{k}^{2}$:
(28)$$\begin{array}{l} J_{m}=\frac{4\pi^{2}f^{2}d^{2}}{\sigma^{2}c^{2}}\left[\begin{array}{l} \begin{array}{cc} \sum_{k=1}^{m}\frac{\left(p_{k}^{4}-2S_{k}^{2}p_{k}^{2}\right)\cos2\varphi_{k}}{S_{k}^{6}}+\frac{1}{S_{k}^{2}} & \sum_{k=1}^{m}\frac{\left(2S_{k}^{2}p_{k}^{2}-p_{k}^{4}\right)\cos\varphi_{k}\sin\varphi_{k}}{S_{k}^{6}} \end{array}\\ \begin{array}{cc} \sum_{k=1}^{m}\frac{\left(2S_{k}^{2}p_{k}^{2}-p_{k}^{4}\right)\cos\varphi_{k}\sin\varphi_{k}}{S_{k}^{6}} & \sum_{k=1}^{m}\frac{\left(p_{k}^{4}-2S_{k}^{2}p_{k}^{2}\right)\sin2\varphi_{k}}{S_{k}^{6}}+\frac{1}{S_{k}^{2}} \end{array} \end{array}\right]\\ =\frac{4\pi^{2}f^{2}d^{2}}{\sigma^{2}c^{2}}\left[\begin{array}{l} \begin{array}{cc} \sum_{k=1}^{m}\frac{\left(p_{k}^{4}-2S_{k}^{2}p_{k}^{2}\right)\left(1+\cos2\varphi_{k}\right)}{2S_{k}^{6}}+\frac{1}{S_{k}^{2}} & \sum_{k=1}^{m}\frac{\left(2S_{k}^{2}p_{k}^{2}-p_{k}^{4}\right)\sin2\varphi_{k}}{2S_{k}^{6}} \end{array}\\ \begin{array}{cc} \sum_{k=1}^{m}\frac{\left(2S_{k}^{2}p_{k}^{2}-p_{k}^{4}\right)\sin2\varphi_{k}}{2S_{k}^{6}} & \sum_{k=1}^{m}\frac{\left(p_{k}^{4}-2S_{k}^{2}p_{k}^{2}\right)\left(1-\cos2\varphi_{k}\right)}{2S_{k}^{6}}+\frac{1}{S_{k}^{2}} \end{array} \end{array}\right], \end{array}$$
whose determinant is
(29)$$\begin{array}{ll} \det\left(J_{m}\right) & ={\left(\frac{4\pi^{2}f^{2}d^{2}}{\sigma^{2}c^{2}}\right)}^{2}\left[{\left(\sum_{k=1}^{m}\frac{p_{k}^{4}-2S_{k}^{2}p_{k}^{2}+2S_{k}^{4}}{2S_{k}^{6}}\right)}^{2}-{\left(\sum_{k=1}^{m}\frac{\left(2S_{k}^{2}p_{k}^{2}-p_{k}^{4}\right)\sin2\varphi_{k}}{2S_{k}^{6}}\right)}^{2}\right]\\ & -{\left(\frac{4\pi^{2}f^{2}d^{2}}{\sigma^{2}c^{2}}\right)}^{2}\left[{\left(\sum_{k=1}^{m}\frac{\left(p_{k}^{4}-2S_{k}^{2}p_{k}^{2}\right)\cos2\varphi_{k}}{2S_{k}^{6}}\right)}^{2}\right]. \end{array}$$

Let $A_{k}=\frac{p_{k}^{4}-2S_{k}^{2}p_{k}^{2}}{2S_{k}^{6}}$; then, we have
(30)$$\begin{array}{l} {\left(\sum_{k=1}^{m}\frac{\left(2S_{k}^{2}p_{k}^{2}-p_{k}^{4}\right)\sin2\varphi_{k}}{2S_{k}^{6}}\right)}^{2}+{\left(\sum_{k=1}^{m}\frac{\left(p_{k}^{4}-2S_{k}^{2}p_{k}^{2}\right)\cos2\varphi_{k}}{2S_{k}^{6}}\right)}^{2}\\ ={\left(\sum_{k=1}^{m}A_{k}\sin2\varphi_{k}\right)}^{2}+{\left(\sum_{k=1}^{m}A_{k}\cos2\varphi_{k}\right)}^{2}\\ =\sum_{k=1}^{m}A_{k}^{2}+\sum_{1\leq i<j\leq m}^{m}2A_{i}A_{j}\cos2\varphi_{i}\cos2\varphi_{j}+\sum_{1\leq i<j\leq m}^{m}2A_{i}A_{j}\sin2\varphi_{i}\sin2\varphi_{j}\\ =\sum_{k=1}^{m}A_{k}^{2}+\sum_{1\leq i<j\leq m}^{m}2A_{i}A_{j}\cos2\left(\varphi_{i}-\varphi_{j}\right)\\ =\sum_{k=1}^{m}A_{k}^{2}+\sum_{1\leq i<j\leq m}^{m}2A_{i}A_{j}\left(1-2\sin^{2}\left(\varphi_{i}-\varphi_{j}\right)\right)\\ =\sum_{k=1}^{m}A_{k}^{2}+\sum_{1\leq i<j\leq m}^{m}2A_{i}A_{j}-\sum_{1\leq i<j\leq m}^{m}4A_{i}A_{j}\sin^{2}\left(\varphi_{i}-\varphi_{j}\right)\\ ={\left(\sum_{k=1}^{m}A_{k}\right)}^{2}-\sum_{1\leq i<j\leq m}^{m}4A_{i}A_{j}\sin^{2}\left(\varphi_{i}-\varphi_{j}\right). \end{array}$$


According to Equations (29) and (30), we have
(31)$$\begin{array}{ll} \det\left(J_{m}\right) & ={\left(\frac{4\pi^{2}f^{2}d^{2}}{\sigma^{2}c^{2}}\right)}^{2}\left[{\left(\sum_{k=1}^{m}A_{k}+\frac{1}{S_{k}^{2}}\right)}^{2}-{\left(\sum_{k=1}^{m}A_{k}\right)}^{2}+4\sum_{1\leq i<j\leq m}^{m}A_{i}A_{j}\sin^{2}\left(\varphi_{i}-\varphi_{j}\right)\right]\\ & ={\left(\frac{4\pi^{2}f^{2}d^{2}}{\sigma^{2}c^{2}}\right)}^{2}\left[\left(\sum_{k=1}^{m}\frac{1}{S_{k}^{2}}\right)\left(\sum_{k=1}^{m}2A_{k}+\frac{1}{S_{k}^{2}}\right)+4\sum_{1\leq i<j\leq m}^{m}A_{i}A_{j}\sin^{2}\left(\alpha_{ij}\right)\right], \end{array}$$
where $\alpha_{ij}=\varphi_{i}-\varphi_{j}$ represents the angle between the projection of the two AUVs and the surface auxiliary vessel on the horizontal plane. Let $\frac{z_{k}}{S_{k}}=\sin\gamma_{k}$. Then, Equation (31) can be simplified to
(32)$$\begin{array}{ll} \det\left(J_{m}\right) & ={\left(\frac{4\pi^{2}f^{2}d^{2}}{\sigma^{2}c^{2}}\right)}^{2}\left[\left(\sum_{k=1}^{m}\frac{1}{S_{k}^{2}}\right)\left(\sum_{k=1}^{m}\frac{\sin^{4}\gamma_{k}}{S_{k}^{2}}\right)+4\sum_{1\leq i<j\leq m}^{m}A_{i}A_{j}\sin^{2}\left(\alpha_{ij}\right)\right]\\ & ={\left(\frac{4\pi^{2}f^{2}d^{2}}{\sigma^{2}c^{2}}\right)}^{2}\left[\left(\sum_{k=1}^{m}\frac{\sin^{4}\gamma_{k}}{S_{k}^{2}}\right)+\sum_{1\leq i<j\leq m}^{m}\frac{\left(\sin^{4}\gamma_{i}\sin^{4}\gamma_{j}+1\right)\sin^{2}\alpha_{ij}}{S_{i}^{2}S_{j}^{2}}\right]\\ & +{\left(\frac{4\pi^{2}f^{2}d^{2}}{\sigma^{2}c^{2}}\right)}^{2}\left[\frac{\left(\sin^{4}\gamma_{i}+\sin^{4}\gamma_{j}\right)\cos^{2}\alpha_{ij}}{S_{i}^{2}S_{j}^{2}}\right]. \end{array}$$

### 3.3. Aided Location Analysis Based on Fisher Information Matrix

In this section, we discuss the optimal formation of different numbers of AUVs in the communication range. According to Equation (32), $\det(J_{1})={\left(\frac{4\pi^{2}f^{2}d^{2}}{\sigma^{2}c^{2}}\right)}^{2}\frac{\sin^{4}\gamma_{k}}{S_{1}^{4}}$ when m=1. In order to make $\det(J_{1})$ as large as possible, the best position for the AUV is directly below the USV. When m=2,3,…, the AUV formation can be arranged in a variety of ways. This work only considers the influence of different angles between the AUVs of underwater targets on positioning accuracy. As k∈2,3,…,m, $S_{k}=S_{0}$, and $\gamma_{k}=\gamma_{0}$, Equation (32) can be simplified to:(33)$$\det\left(J_{m}\right)={\left(\frac{4\pi^{2}f^{2}d^{2}}{\sigma^{2}c^{2}}\right)}^{2}\left[3m\frac{\sin^{4}\gamma_{0}}{S_{0}^{4}}+\frac{{\left(\sin^{4}\gamma_{0}+1\right)}^{2}}{S_{0}^{4}}\sum_{1\leq i<j\leq m}^{m}\sin^{2}\left(\alpha_{ij}\right)\right].$$

When the distance between the AUVs and the USV is equal and the AUVs are at the same depth, only the angle of the AUVs is considered, and the influence of AUV formation on positioning accuracy is only related to $\sin^{2}\alpha_{ij}$. After calculation, the projections of the optimal array formations on the horizontal plane when m=2,3,4 are obtained, as shown in Figure 3. Specifically, m=3 has two optimal array formations.

### 3.4. USV Path Planning Based on Dubins Path

After the AUV transmits its motion and attitude information to the USV, the USV ensures the positioning accuracy of the AUV by planning the path based on the Fisher information matrix. In order to save the energy of the USV, the total path length of the USV should be reduced as much as possible in path planning. In terms of positioning accuracy, the horizontal distance $r_{m}=\arg\max\left\{\det\left(J_{m}\right)\right\}$ can be obtained according to Equation (32). In order to reduce noise interference, specific scientific missions require the USV and AUV k to maintain a specific minimum horizontal spacing ${\hat{r}}_{k}$. Based on the minimum turning radius $r_{s}$ (Dubins radius) of the USV, we can map the position of the AUV k as a circle at the surface. The radius $r_{k}=\max\left\{{\hat{r}}_{k},r_{m},r_{s}\right\}$ of the circle is the optimal horizontal distance between the USV and AUV k. For USV path planning with curvature constraints, the Dubins path should be used if the start and end locations are specified [25]. Dubins paths indicate that under the maximum curvature restriction, the shortest feasible path between two oriented points in the plane is the CLC path or the CCC path or a subset of them, where C stands for arc segment and L for line segment. This work considers CLC-type paths. The path planning algorithm is divided into two parts: path generation and path finding. The former is used to directly generate the Dubins path and check whether the radius requirements are met. If the radius requirements are not met, the intermediate path is generated according to the tangent line of the circle, and the intermediate path is checked until the radius limit is met.

For example, Figure 4 shows the path of the USV on the surface plane. O is the current position of the USV, $T_{k}$ is the projection of the AUV k on the surface, $T_{d}$ is the target position of the USV, and $\overset{⏜}{OA}-AB-\overset{⏜}{BT_{d}}$ is the path of the USV. The center angles of $\overset{⏜}{OA}$ and $\overset{⏜}{BT_{d}}$ are $\varphi_{1}$ and $\varphi_{2}$, respectively. If the heading angles of the USV and AUV are known to be δ and β, respectively, the USV velocity direction is taken as the positive direction of the Y-axis. The starting point O is the origin, and the coordinate of $T_{k}$ is $\left(x_{k},y_{k}\right)$. Then, the coordinates of each key point can be obtained:(34)$$P_{A}=[r_{1}-r_{1}\cos\varphi_{1},r_{1}\sin\varphi_{1}],$$
(35)$$P_{T_{d}}=[x_{3}-r_{3}\cos(\beta -\delta ),y_{3}-r_{3}\sin(\beta -\delta )],$$
(36)$$\begin{array}{l} x_{B}=x_{3}-(r_{3}+r_{2})\cos(\delta -\beta )+r_{2}\cos(\varphi_{2}+\delta -\beta ),\\ y_{B}=y_{3}-(r_{3}+r_{2})\sin(\delta -\beta )-r_{2}\sin(\varphi_{2}+\delta -\beta ). \end{array}$$

If the AB line is the common tangent of the two arcs, then the slope of AB meets:(37)$$\frac{y_{3}-(r_{3}+r_{2})\sin(\delta -\beta )-r_{2}\sin(\varphi_{2}+\delta -\beta )-r_{1}\sin(\varphi_{1})}{x_{3}-(r_{3}+r_{2})\cos(\delta -\beta )+r_{2}\cos(\varphi_{2}+\delta -\beta )-r_{1}+r_{1}\cos(\varphi_{1})}=\frac{1}{\tan(\varphi_{1})}.$$

**Figure 4 sensors-23-07429-f004:**
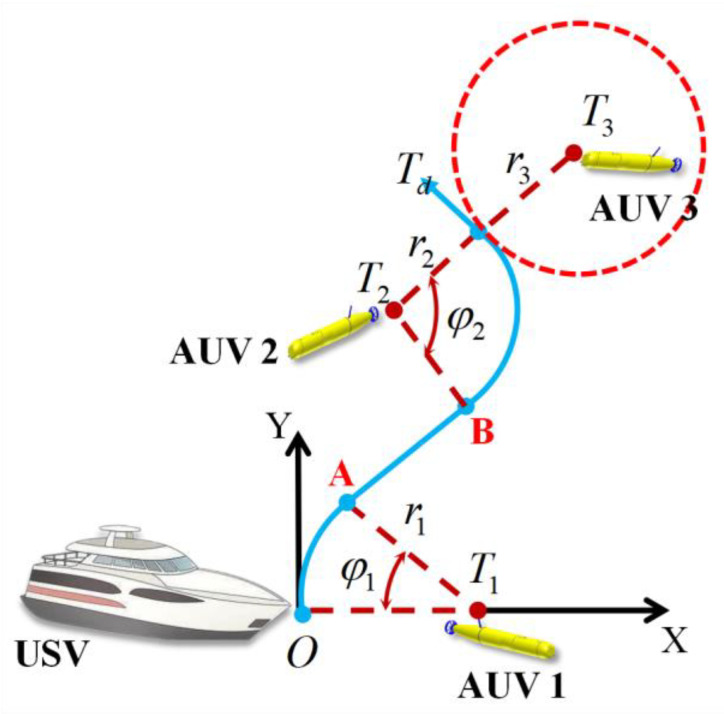
Schematic diagram of USV path planning based on Dubins paths.

As we have $\varphi_{2}-\left(\beta -\delta \right)=\varphi_{1}$, we can denote $C=y_{3}-(r_{3}+r_{2})\sin\left(\beta -\delta \right)$, $D=x_{3}-(r_{3}+r_{2})\cos\left(\beta -\delta \right)-r_{1}$, and tanξ=D/C. Thus, $\sin\left(\varphi_{1}-\xi \right)=\frac{r_{1}+r_{2}}{\sqrt{C^{2}+D^{2}}}$, and we can obtain $\varphi_{1}$. Moreover, the entire path length of the moving USV is $\varphi_{1}r_{1}+\varphi_{2}r_{2}+\frac{C+\sin\left(\varphi_{1}\right)(r_{1}+r_{2})}{\cos\left(\varphi_{1}\right)}$. Then, we can choose the appropriate path according to the length of the path as the location path for the USV.

## 4. Simulation Settings and Results

In this study, we considered the AUV formation for simulation verification in a 3000 m×3000 m×1000 m underwater environment. The maximum number of AUVs in a formation is 4. The underwater sound velocity is considered to be 1500 m/s. The array space is 4 cm, and the Dubins radius of the USV is 3 m. The source level of the sonar is TL=500 dB, the return signal strength is TS=3 dB, the communication threshold is DT=10 dB, the center frequency of the sonar is f=10 kHz, and the number of transducer units is $n_{tr}=5$. As for the underwater environment, the wind speed is w=0.2 m/s and the shipping activity is ς=0.5. The above parameter settings are referenced and based on [20,21,22,26,27,28,29,30]. We then simulated and analyzed the proposed scheme based on the MATLAB platform.

### 4.1. Path Planning for the Moving USV

In the process of optimizing positioning accuracy with the Fisher information matrix, the USV needs to adjust its position according to the position of the AUVs. Let $p^{2}=x^{2}+y^{2}$, and we can analyze the magnitude of the determinant $\det(J_{m})$ of the Fisher matrix in the vertical plane of the USV and AUVs with respect to horizontal distance and depth. In Figure 5, our simulation verifies the relationship between $\mathrm{lg}\left(\det(J_{m})\right)$ and z, p in different AUV formations. The USV corresponds to the origin in the upper left corner of the figure, and the positioning accuracy of the AUV at different positions in the figure is presented in the form of contour lines. With a given AUV depth, the relative position range of the USV and AUV can be obtained when the positioning accuracy is high. The range obtained can be used for subsequent Dubins path planning. In practice, each AUV needs to make a circular path in this area to perform certain scientific tasks. Then, the USV routing algorithm abstracts AUV k into circular obstacles according to $r_{k}$, and it performs intermediate pose generation and path connection on the original Dubins path.

Specifically, the initial pose information $\left(x=20\;m,\;y=0\;m,\;\alpha =60{}^{\circ}\right)$ and the final target pose information $\left(x=45\;m,\;y=20\;m,\;\alpha =60{}^{\circ}\right)$ of the USV are given. Assuming that there are three unlocated AUVs, the corresponding coordinate information and limit distance are $\left(x_{1}=25\;m,\;y_{1}=10\;m,\;{\hat{r}}_{1}=3\;m\right)$, $\left(x_{2}=25\;m,\;y_{2}=23\;m,\;{\hat{r}}_{2}=4\;m\right)$, and $\left(x_{3}=36\;m,\;y_{3}=40\;m,\;{\hat{r}}_{3}=3\;m\right)$. The USV path planning based on the proposed scheme can be seen in Figure 6. When there is more than one Dubins circle between the starting point and the destination point, there is more than one Dubins path. In this case, many different paths can be calculated, and when comparing the lengths of the all paths, the shorter path can be regarded as the better path, that is, to find the best path among all the combinations of paths that meet the conditions. As can be seen, we can make the USV approach the optimal distance of each AUV for the subsequent underwater positioning. By choosing the right path, we can reduce the total path length of the USV.

### 4.2. Analysis of Distance to Positioning Accuracy

The point error is introduced to analyze the positioning accuracy, that is, the sum of the variance of the X direction and the Y direction of the AUV’s positioning $\varepsilon_{p}^{2}=\varepsilon_{x}^{2}+\varepsilon_{y}^{2}$. In practice, the variance of sequence x and y coordinates is obtained via repeated independent sampling of the AUV’s positions under exactly the same conditions. Monte Carlo sampling is taken, where the number of repetitions is uniformly 500. All position-related data are obtained using the USBL algorithm that simulates the USV and AUVs.

In Figure 7, we show the changes in point error and Fisher information with the horizontal distance between the AUV and USV, when the formations with different numbers of AUVs adopt the optimal array. It can be seen that with the increase in horizontal distance, the point error increases and the Fisher information decreases, indicating that under the optimal array, the closer the distance between the AUV and the USV, the higher the positioning accuracy.

### 4.3. Analysis of Formation to Positioning Accuracy

When the number of AUVs in the formation is 3, $\alpha_{12}=\alpha_{23}=\theta$ is used as the parameter to describe the array formation. When m=3, the positioning accuracy based on Fisher information varies with horizontal distance p and the angle θ, as shown in Figure 8. The experimental depth of AUVs was set at 800 m. It can be seen that the greater the ratio of horizontal distance to depth, the greater the impact of formation on positioning accuracy. When m=3, the best formation appears at 60° and 120°, which is consistent with the results of theoretical derivation.

When m=4, set $\alpha_{12}=\alpha_{34}=90{}^{\circ}$ and use $\alpha_{13}=\theta$ to describe the parameters of array formation. When m=4, the positioning accuracy based on Fisher information changes with horizontal distance p and angle θ, as shown in Figure 9. The experimental depth was set at 800 m, and the conclusion was the same as m=3. The larger the ratio of horizontal distance to depth, the greater the influence of array formation on positioning accuracy. Since θ=0° does not meet the definition, the optimal formation occurs at 90°, which is consistent with the theoretical derivation. In addition, the benefit of changing the formation of m=4 is smaller than that of m=3, which is speculated to be related to the increase in the total number of AUVs and the increase in the information available to the USV to help eliminate positioning errors.

## 5. Conclusions

When multiple AUVs perform similar underwater tasks, such as conducting scientific research in a certain area, the AUVs often need to form a formation to maintain a similar trajectory and move uniformly; they also need to confirm the position of each AUV in real time. At this time, if a USV equipped with GPS is introduced to follow the AUV formation, and each AUV can communicate with it, the high-precision positioning of each AUV can be realized. Based on the above background, we constructed a new surface–underwater cooperative positioning system based on underwater acoustic communication, which is called the USV–AUV network. Specifically, we proposed an orthogonal array USBL cooperative location scheme based on sonar carried by the USV. Using the derivation of the Fisher information matrix formula under Cartesian parameters, we analyzed the positioning accuracy of AUVs in different positions under the USBL positioning mode and gave the optimal array for the AUV formation. Additionally, we introduced Dubins path planning functions for the design of the USV path planning algorithm. Finally, a large number of simulation experiments verified the performance of the proposed scheme in AUV positioning, which can effectively help underwater research missions based on the AUV formation.

## Figures and Tables

**Figure 1 sensors-23-07429-f001:**
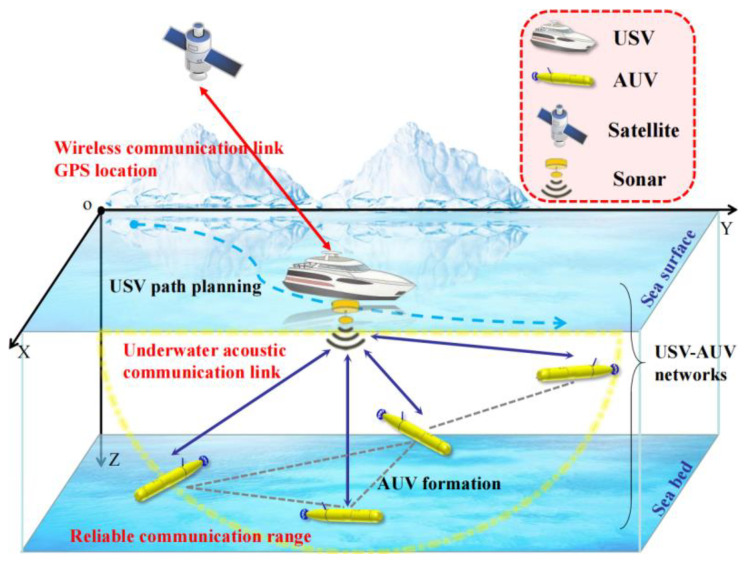
Illustration of USV–AUV networks.

**Figure 2 sensors-23-07429-f002:**
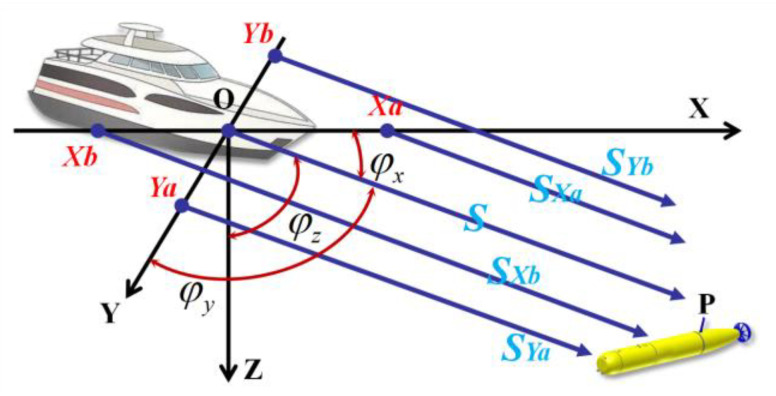
Schematic diagram of the orthogonal array USBL cooperative location.

**Figure 3 sensors-23-07429-f003:**
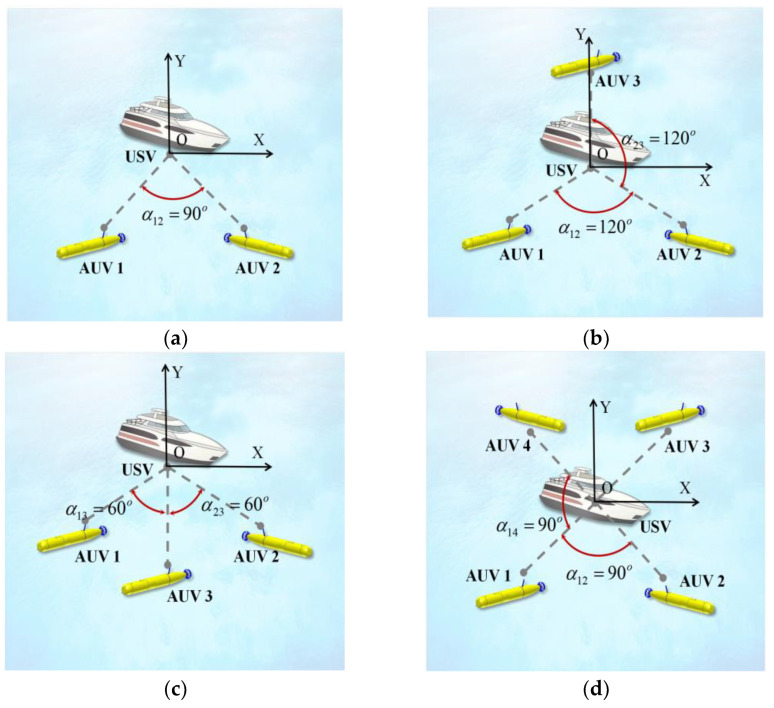
Optimal arrays for different numbers of AUVs. (**a**) m=2. (**b**) First case, where m=3. (**c**) Second case, where m=3 and (**d**) m=4.

**Figure 5 sensors-23-07429-f005:**
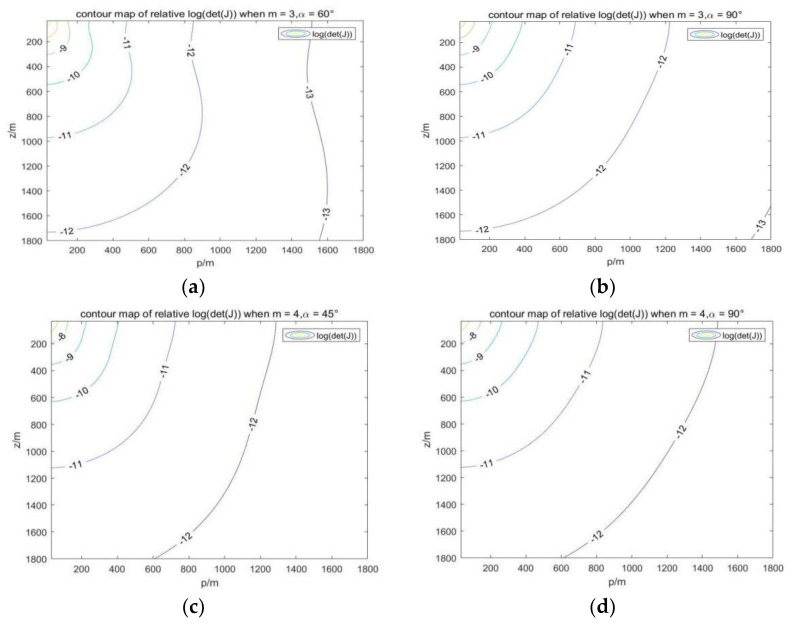
Change in determinant size with z and p in different AUV formations. (**a**) m=3,α=60°. (**b**) m=3,α=90°. (**c**) m=4,α=45°. (**d**) m=4,α=90°.

**Figure 6 sensors-23-07429-f006:**
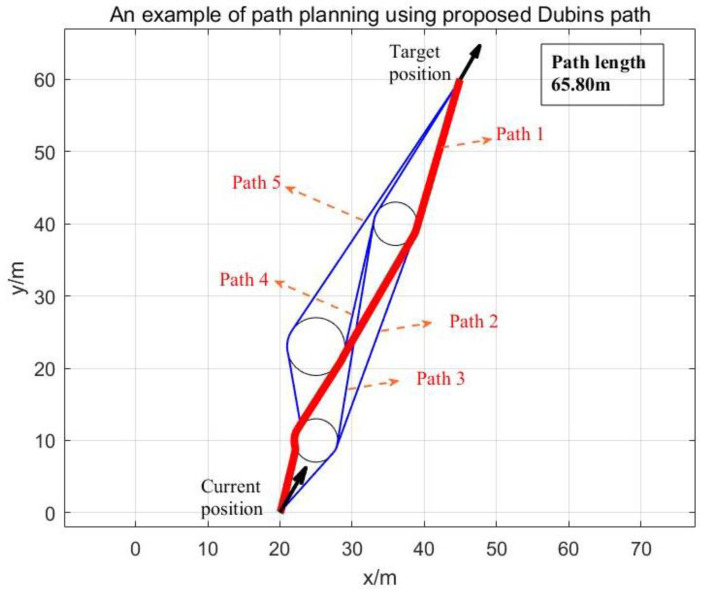
Diagram of the USV path planning.

**Figure 7 sensors-23-07429-f007:**
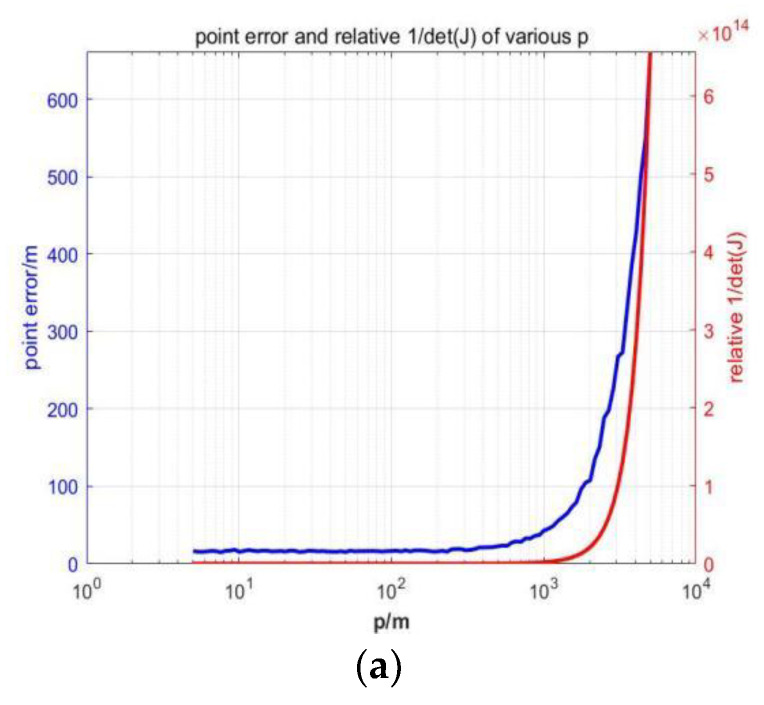
Changes in point error and Fisher information with the horizontal distance between the AUV and USV. (**a**) m=2. (**b**) m=3.

**Figure 8 sensors-23-07429-f008:**
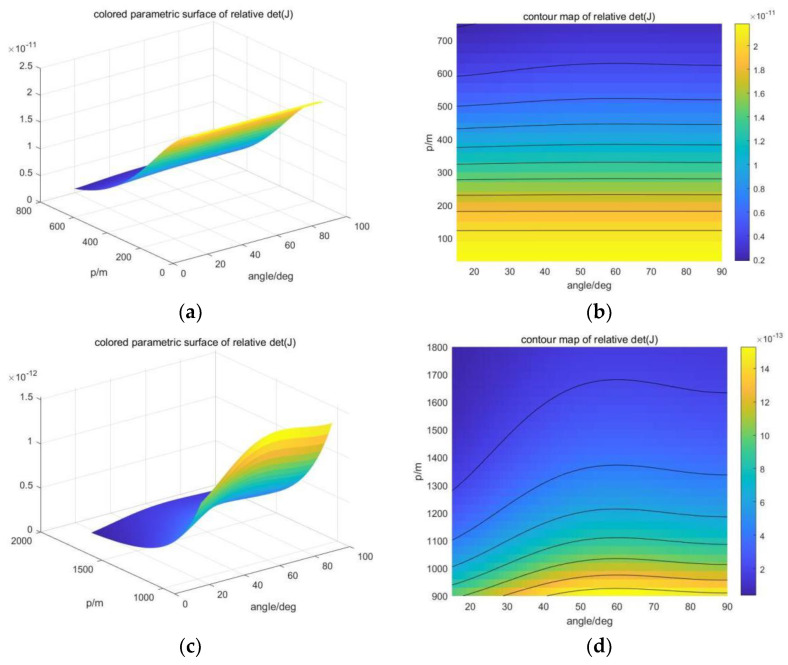
Positioning accuracy based on Fisher information varies with horizontal distances and angle when m=3. (**a**) Three-dimensional closer case. (**b**) Two-dimensional closer case. (**c**) Three-dimensional far case. (**d**) Two-dimensional far case.

**Figure 9 sensors-23-07429-f009:**
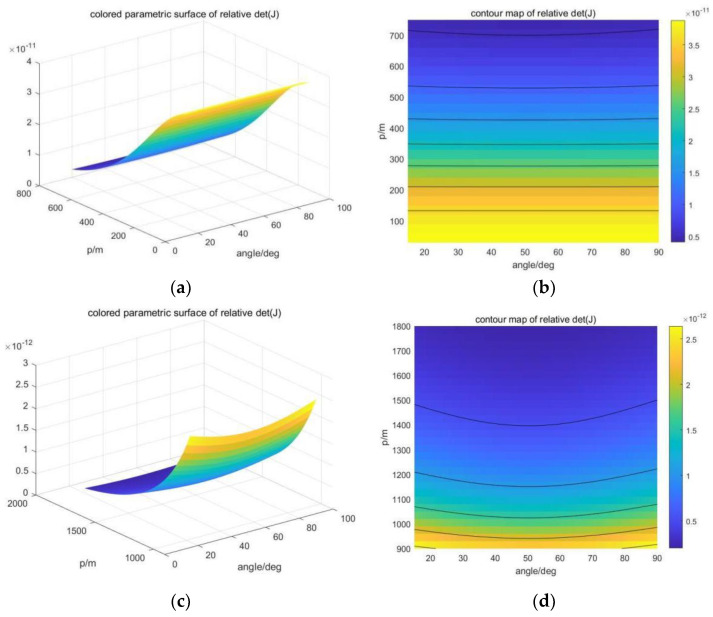
Positioning accuracy based on Fisher information varies with horizontal distances and angle when m=4. (**a**) Three-dimensional closer case. (**b**) Two-dimensional closer case. (**c**) Three-dimensional far case. (**d**) Two-dimensional far case.

## Data Availability

The data that support the findings of this study are available from the corresponding author, [Men, W.], upon reasonable request.

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
