# Peer review of "Fisher-Information-Matrix-Based USBL Cooperative Location in USV–AUV Networks"

_sensors, 2023, doi:10.3390/s23177429_

Round 1

Reviewer 1 Report

# Review Report on "Fisher Information Matrix Based USBL Cooperative Location in USV-AUV Networks"
## Summary
This manuscript proposes a USV-AUV network and a USBL acoustic positioning scheme to locate the AUV. The positioning accuracy of AUVs in different positions was analyzed using the Fisher information matrix. A USV path planning scheme is proposed to save the energy of USV and assist in locating the AUV. The manuscript's logical flow is clear, and it provides rich content. However, there are some issues that need improvement, which are summarized as follows.
## Major Comments
1. Abbreviations Definition: The manuscript defines some terms, such as USV, USBL, and AUV, multiple times. It is recommended to define these abbreviations at the beginning and use them consistently throughout the manuscript.
2. Figure 3 - Coordinate System: The coordinate system in Figure 3 is not clearly defined. It is suggested to include coordinate axes in the figure to provide clarity.
3. Figure 6 - Color and Number Meaning: The meaning of colors and numbers in Figure 6 is unclear, and the caption lacks sufficient clarity. Adding a clear explanation for the colors and numbers will enhance the figure's understanding.
4. Figure 7 - Y-Axis Color and Label Clarity: In Figure 7, the blue line corresponds to the left y-axis, and the red line corresponds to the right y-axis. However, the color of the y-axes does not match the line color. To improve visual consistency, it is recommended to change the y-axes colors to blue and red. Additionally, the figure appears cropped, making the x-axis label unclear.
5. Equation Typesetting: The equations throughout the manuscript appear much larger than the body text, making them visually distracting and disrupting the flow of reading. The equations in lines 269-285 are particularly poorly typed, with some superscripts not appearing correctly. The manuscript needs careful proofreading and reformatting to ensure that all equations are correctly typed and appropriately sized to match the body text.
## Recommendations
I strongly urge the authors to thoroughly review the manuscript to address equation typesetting issues. Ensuring proper formatting and consistency with the body text will greatly enhance the presentation and readability of the research. Resolving the mentioned issues will significantly improve the manuscript's clarity and overall quality.

N/A

Reviewer 2 Report

The authors proposed a topic of Fisher Information Matrix Based USBL Cooperative Location in 2 USV-AUV Networks using simulation tool. Authors have to fix  the following points.

1. The detail of section 5 is missing in section 1 (Introduction), also in the manuscript.

2. Typo in line 129 (equation 3a), and 133 (equation 3d), equations are not properly aligned.

3. The angles (in Phi) mentioned in line 152, and 153 are not same as mentioned in figure 2 (in Theta). Also, the caption is not with the figure 2 (Caption should be on the same page with figure).

4. There is a typo in the caption of figure 3.

5. The quality of figures in result section is not good, please try to improve.

6. The coordinates (USV path positions) overlapping in figure 6.

7. 

Reviewer 3 Report

Good effort

Minor edits needed
